

# A frog in hot water: the effect of temperature elevation on the adrenal stress response of an African amphibian

Juan Scheun[1,2], Leanne Venter[1] and Andre Ganswindt[2]

[1] Department Nature Conservation, Faculty of Science, Tshwane University of Pretoria, Pretoria, Gauteng, South Africa
[2] Mammal Research Institute, Department Zoology and Entomology, University of Pretoria, Pretoria, Gauteng, South Africa

## ABSTRACT

Amphibians, with their unique physiology and habitat requirements, are especially vulnerable to changes in environmental temperatures. While the activation of the physiological stress response can help to mitigate the impact of such habitat alteration, chronic production of elevated glucocorticoid levels can be deleterious in nature. There is no empirical evidence indicating the physiological response of African amphibians to temperature changes, where individuals are unable to emigrate away from potential stressors. To rectify this, we used the edible bullfrog (*Pyxicephalus edulis*) as a model species to determine the effect of elevated temperature on the adrenocortical response of the species using a recently established matrix. While a control group was kept at a constant temperature (25 °C) throughout the study period, an experimental group was exposed to control (25 °C) and elevated temperatures (30 °C). Mucous swabs were collected throughout the study period to determine dermal glucocorticoid (dGC) concentrations, as a proxy for physiological stress. In addition to this, individual body mass measurements were collected. The results showed that individuals within the experimental group who experienced increased temperatures had significantly elevated dGC levels compared to the control animals. Furthermore, there was a significant difference in the percentage mass change between experimental and control animals . These findings indicate the physiological sensitivity of the edible bullfrog to a thermal stressor in captivity. While this study shows the importance of proper amphibian management within the captive environment, it also highlights the coming danger of global climate change to this and similar amphibian species.

# INTRODUCTION

Temperature stands out as a crucial factor influencing the development and survival of herptiles (*Tattersall et al., 2012*; *White, Alton & Frappell, 2012*). The physiological and biological functions of reptiles and amphibians have evolved in close correlation with temperatures specific to their habitats (*Khelifa et al., 2019*). While ectotherms thrive in specific, often elevated temperatures (referred to as the "optimal temperature"), exposure to temperature extremes beyond an individual's tolerance range can induce substantial

Corresponding author
Juan Scheun, scheunj@tut.ac.za

alterations in biochemical processes, individual fitness, and overall health (*Gunderson & Stillman, 2015*; *Huey & Kingsolver, 1989*). Among vertebrates, amphibians emerge as particularly susceptible to temperature variability. This vulnerability is attributed to their distinctive physiology, specific habitat requirements, and a limited capacity to migrate over substantial distances in pursuit of new habitats (*Baldwin, Calhoun & De Maynadier, 2006*; *Carvalho, Navas & Pereira, 2010*; *Foden et al., 2013*). Consequently, amphibians exhibit a diminished capacity to respond as efficiently to drastic temperature fluctuations compared to other vertebrate wildlife (migration/physiological adaptation/phenotypic plasticity; *Bernardo & Spotila, 2006*; *Kearney, Shine & Porter, 2009*). These factors, along with the low levels of heat tolerance inherent in amphibians (*Nowakowski et al., 2018*), means a sudden increase in ambient temperature can result in distinct changes in physiological and biochemical processes (*Carey & Alexander, 2003*). In addition to this, elevated temperatures can also lead to dehydration and lowered survival rates of amphibian species (*Lertzman-Lepofsky et al., 2020*).

Numerous studies have attempted to elucidate the impact of temperature variability on free-ranging amphibian populations through the application of direct monitoring or modelling tools (*Bartelt & Peterson, 2005*; *Enriquez-Urzelai et al., 2019*; *Winter et al., 2016*). Despite the efficacy of these tools, their implementation in the captive environment has been limited. This is strange, as by 2020 more than 540 amphibian species were captively housed, totalling 7% of the 7,658 known amphibian species (*Jacken, Rödder & Ziegler, 2020*). While many amphibians are kept legally in captive environments such as zoos for display purposes, it is important that housing conditions are sufficient to ensure individual health and welfare (*Nash, 2005*). Furthermore, despite an ever-growing call to enhance the ability to monitor the health and welfare of captive animals (*Ferrie et al., 2014*), very few facilities housing amphibians have incorporated robust behavioural and physiological monitoring programs to ensure captive individuals will be able to survive current threats (*Harding, Griffiths & Pavajeau, 2016*). Ensuring the effective and optimal temperature regime within the captive environment is a key factor which needs to be determined for amphibians (*Carrillo, Johnson & Mendelson III, 2015*). Importantly, this includes an understanding of a species' temperature tolerance range, as temperatures reaching the upper or lower limit of an organism may result in increased physiological stress. As an elevation in temperature can activate the physiological stress response in many wildlife species (*Alfonso, Gesto & Sadoul, 2021*; *Frigerio et al., 2004*), therefore implementing endocrine monitoring may be an ideal tool for monitoring the response of amphibians to management practices, such as temperature determination, in captivity.

The physiology stress response includes the hyperactivation of the hypothalamic-pituitary-interrenal (HPI) axis and an increase in secreted glucocorticoids (GC). An acute elevation in GC levels is adaptive in nature, ensuring an individual can survive within a challenging environment by altering metabolic function and behaviour (*Sapolsky, Romero & Munck, 2000*; *Wack et al., 2012*). However, chronically elevated GC levels can be detrimental to amphibians, leading to a decrease in adaptability and coping capacity against natural and anthropogenic stressors (*Baker, Gobush & Vynne, 2013*; *Boonstra et al., 2001*; *Narayan & Hero, 2014*; *Phillips & Puschendorf, 2013*). As a result of the importance of GCs

in amphibian welfare, the quantification of adrenocortical hormones in amphibians under thermal stress, in both controlled and field conditions, can provide valuable information on the "sublethal" effects of temperature fluctuations in these animals (*Narayan, 2016*).

Endocrine monitoring has primarily been conducted through the collection and analysis of blood or excreted material such as urine and faeces (*Sheriff et al., 2011*). However, both techniques have considerable flaws when implemented in amphibian research. Firstly, to collect blood samples, animals need to be captured and restrained for prolonged periods at a time, resulting in additional stress for the animal and a possible GC feedback (*Romero & Reed, 2005*; *Tylan et al., 2020*). In addition to this, the small body size of most amphibians results in limited venipuncture sites (*Heatley & Johnson, 2009*; *Tapley, Acosta-Galvis & Lopez, 2011*). Although urine and faecal collection, and the analysis of hormone metabolites therein, is considered non-invasive in nature (*Hodges, Brown & Heistermann, 2010*), the small body size inherent in many amphibian species can result in the collection of insufficient sample mass; furthermore, the extended gut passage time in many species (>33 h; *Jiang & Claussen, 1993*) inhibits the ability to collect frequently from monitored populations (*De la Navarre, 2006*). A recently validated method for monitoring GC levels in the dermal secretion of amphibians was developed by *Santymire, Manjerovic & Sacerdote-Velat (2018)*. Although the technique is semi-invasive in nature, requiring the brief capture and restraint of study animals, the process ensures longitudinal sampling, while avoiding the effect of GC feedback. Similar to urine and faeces, dermal GC (dGC) concentrations represent an integrative measure of GCs across a prolonged period, *e.g.*, "pooled" (*Scheun et al., 2019*). Despite the numerous advantages of using mucous as a monitoring matrix, no study to date has used it to monitor the physiological stress response in any amphibian species in response to elevated ambient temperatures. However, studies monitoring amphibian urinary glucocorticoid metabolite (uGCM) concentrations in response to temperature elevation found a significant increase in uGCM levels following temperature alterations (*Jessop et al., 2018*).

To monitor the effect of elevated temperature on the adrenocortical response in amphibians, we used the edible bullfrog (*Pyxicephalus edulis*) as a study species. The edible bullfrog is an African amphibian found throughout sub-Saharan Africa, where the species lives underground until habitat conditions improve following seasonally dependent rainfall (*Jacobsen, 1989*; *Vlok et al., 2013*). The genus *Pyxichephalus* are known to enter belowground sites and hibernate to avoid unfavourable conditions (*Thomas et al., 2014*); individuals will emerge from their belowground sites when rainfall occurs, usually in spring or summer (*Channing, Du Preez & Passmore, 1994*). The commonly held believe that species capable of retreating underground to escape unfavourable conditions are thereby better equipped to withstand such events may not be entirely accurate. As extreme weather events become more intense and frequent (*Stott, 2016*), initial models suggest that temperatures below ground could rise by 5 °C by 2065 (*Petrie et al., 2020*). Further projections indicate that an increase of more than 10 °C at depths of 0–20 cm below the surface can is expected beyond 2065. Thus, underground behaviours displayed by amphibian species will likely not protect individuals from climate change. Except for the validation of mucous and urine as robust matrices for monitoring GCs and their
metabolites in the species (*Scheun et al., 2019*), no further endocrine research has been conducted on the species. As such, the current study aimed to increase our understanding of the physiological responses through the HPI axis and body mass in an African amphibian to elevated temperature change over prolonged periods (5 weeks).

We hypothesised that an increase in environmental temperatures would lead to the hyperactivation of the HPI axis, resulting in a significant increase in GC secretion in dermal glucocorticoid concentrations. Similarly, we hypothesised that an increase in environmental temperature will lead to a decrease in body mass.

## MATERIALS & METHODS

### Study animals and site

The study was conducted on fourteen edible bullfrogs between 3 December 2019 and 11 February at the National Zoological Garden (NZG), South African National Biodiversity Institute, Pretoria, South Africa (25.7469°S, 28.18918°E). All study animals were housed at the NZG as part of the animal collection since being confiscated from poachers collecting individuals within the wild in 2016 and are thus defined as adult. Due to limited information on where the poachers collected individuals, all confiscated animals could not be released into the wild. The study species was selected due to their availability at the study site (NZG), as well as their general distribution throughout Africa, making them an ideal model species. The identification of sex is difficult within the species, as only a slight size difference between females and males is evident (*Braack & Maguire, 2005*). Like many amphibian species, the edible bullfrog has a tougher dorsal region, which minimises evaporative water loss, with the majority of water loss and absorption occurring through the ventral regions (*Jørgensen, 1997*; *Lillywhite, 2006*). The species is currently listed as least concern by the IUCN Red List of Threatened Species (*IUCN SSC Amphibian Specialist Group, 2016*). The sample size was determined by the number of available individuals. Criteria for excluding individuals were set; all individual deemed unhealthy by the chief veterinarian at the NZG would be excluded from the study. All individuals were considered healthy and suitable for the purpose of the study following a health check. Upon arrival at the NZG all individuals were tested for *Batrachochytrium dendrobatidis*; all tests for the pathogen return a negative result. As the level of sexual dimorphism is limited in the species, no distinction between male and female could be made during this study. Prior to this study, all individuals were housed individually in transparent plastic containers (30 cm × 25 cm × 20 cm) between 23–25 °C. At the start of the study, animals were moved to a cordoned off section of the Reptile and Amphibian section and placed into new containers containing a layer of clean, non-chlorinated water (24 °C) originating from the water purification system at the Reptile and Amphibian section. When clean water was needed, a researcher would empty the container and add 750 ml of water (one cm in height). The water in each container was replaced weekly to avoid dirty/murky water or excessive evaporation. Each container had an elevated platform which allowed an individual to exit the water and a dark, circular container in which the individual could take refuge but not escape the environmental temperatures. Individuals were kept on a natural light-dark cycle, linked

to naturally occurring sunrise and sunset during the study period (December–February 14:16 h–8:10 h). Individuals were fed a range of insects and/or rodents every 2–3 days as set out by the NZG Reptile and Amphibian section staff. The amount of food provided to individuals were not altered throughout the study period.

All study animals were allowed to acclimatize to their new enclosure for 30 days at 25 °C prior to the onset of the study. To ensure that temperatures remained constant throughout the study, temperatures were monitored within the containers as well as throughout the cordoned off study sections using universal digital thermometers with a 2 m cable (Electromann SA, South Africa). Following this, individuals were randomly divided into control ($n = 7$) and experimental ($n = 7$) groups. To ensure individuals were randomly selected to either the control or experiment group, each individual was assigned a number. Only the primary investigator (Scheun, J) was aware of group allocations throughout the study period, while every effort was made to keep this information removed from the laboratory technicians and co-investigators until manuscript preparation. These numbers were randomised in excel and the first seven individuals placed into the control group, while the remaining seven were placed into the experiment group.

Experimental animals were kept at 25 °C ± 0.6 standard deviation (SD) for five weeks (Week 1–5). Following this, individuals belonging to the experimental group were kept at a temperature of 30 °C ± 0.8 SD for a further five weeks (week 6–10). Animals within the control group were kept at 25 °C ± 0.5 SD throughout the first and second states (week 1–5, 6–10). For monitoring dGC levels in experimental animals, states were named "BaselineEXP" (Week 1–5) and "Elevated" (week 6–10); similarly, states were referred to as "Baseline Ctr" (Week 1–5) and "Baseline Ctr 2" (week 6–10) for control animals. All animals were weighed at three points throughout the study: At the start of the study (week 1) , at the end of the baseline period for both group (week 5) and at the end of the study (week 10).

As all individuals were habituated to the 24–25 °C range prior to, and during the acclimatisation period, the 5 °C increase used in this study would signal a prolonged, extreme event in the natural environment. The prolonged temperature elevation was conducted to ensure that the physiological and physical response of each experimental animal could be observed.

Temperature of both control and experimental groups were maintained throughout the day (06:00–18:00) using heat lamps, which were switched off at night (18:00–06:00) to simulate nature light and temperature cycles. Nighttime temperatures in the cordoned off section ranged from 15.8–19.9 °C as noted by the Reptile and Amphibian section's temperature recorders. Sampling order was randomized in excel to ensure no individual was repeated sampled at the same time point during sampling events. Dermal swabs were collected weekly, to reduce animal discomfort and stress, throughout the study based on the technique described by *Scheun et al. (2019)*. In short, researchers handled individuals with fresh, disposable gloves, taking care not to make contact with the dorsal region. Individuals were held in one hand and gently, but firmly swabbed (2 mm-diameter plastic cotton swabs without adhesive; CitoswabR transport swab, 2120-0015, Haimen City, China) three times across the dorsal region along a length of approximately 2.5 cm. Sampled individuals
were placed back into their respective enclosure. Swabs containing mucous were placed into a 2 ml microcentrifuge tube containing 70% ethanol and sealed with parafilm to reduce evaporative loss and potential leakage, before being stored at −20 °C until sample extraction.

The study was approved by the SANBI National Zoological Garden Animal Use and Care Committee (Reference: P19/15). At the end of the study period all study animals ($n = 14$) were returned to the species-approved housing provided by the Reptile and Amphibian section of the NZG. As with during the study period, animal care and welfare was managed by the staff compliment of the Reptile and Amphibian section upon the completion of the study. Humane endpoints were included in the study: Individuals showing a loss of appetite and lethargic behaviour would be removed from the study. No individuals met this criterion throughout the study period.

## Sample extraction & Enzyme immunoassay analysis

Dermal secretion samples were extracted according to a process developed by *Santymire, Manjerovic & Sacerdote-Velat (2018)* and validated for the study species (*Scheun et al., 2019*). As developed by *Santymire, Manjerovic & Sacerdote-Velat (2018)*, samples were kept at room temperature for 30 min prior to the start of the extraction process. Samples were shaken in a water bath shaker at 70 rpm for 5 min before being briefly vortexed. Following a 15 s centrifuge spin down (1,500 g), 0.5 ml of each sample was removed and placed into a new, pre-labelled 2 ml microcentrifuge tube. The extracts were then placed into an incubator oven at 60 °C until dry (∼5 h). Three glass beads were added to each tube containing dried extract prior to the addition of 0.5 ml assay buffer. Samples were then vortexed for 15 s before being placed into a sonicator for 20 min. Finally, the samples were shaken on a water bath shaker for 30 min at 70 rpm. Sample extracts were stored at −20 °C until enzyme immunoassay (EIA) analysis.

Dermal glucocorticoid concentrations were determined in extracts using a corticosterone EIA previously validated for the species by *Scheun et al. (2019)*. Details of the respective EIA, including cross-reactivities, are given by *Touma et al. (2003)*. Assay sensitivity was 0.08 ng/ml. Intra- and inter-assay coefficient of variance, determined by repeated measurements of high- and low-quality controls, were 4.57% and 5.74%, as well as 5.43% and 8.07%, respectively. As samples were analysed neat, and thus in the same dilution, no parallelism test for the corticosterone EIA was conducted.

## Data analysis

All statistical analyses were conducted using R (*R Core Team, 2019*). Significance was set at $p < 0.05$.

## Dermal glucocorticoid concentrations

No criteria were set for the exclusion of data points. For data analyses, all data points for control ($n = 7$) and experiment ($n = 7$) animals were used. For the 140 dermal secretion samples analysed, the dependent variable (dGC) was not normally distributed (Shapiro–Wilk, $W = 0.971$, $p < 0.005$).

Due to the non-normal distribution of dGC, log transformation was applied to the data to meet the assumptions required for parametric statistical analysis. Log transformation (natural logarithm using Euler's number 'e') has been used in several studies, many of which highlighting the importance of transforming endocrine and other biological data in such a way (*Miller & Plessow, 2013*; *Stewart, 2015*). The transformation was performed according to the formula dGC_log = log10(dGC), where dGC represents the original dGC measurements. As included dGC values were below 1, log10 transformed values are given as negative values. Individual ID was included as a random effect to account for the intra-individual variability and to control for potential non-independence of observations collected from the same individuals. This approach allows us to generalize the effects of other explanatory variables across the population, while accounting for the repeated measures within subjects. Following this process dGC data were reanalysed using a Shapiro–Wilk test; the dGC_log data were found to be normally distributed ($W = 0,992$, $p$-value $= 0.575$) and could be used for parametric statistical analyses. All graphical representations were subsequently created using log-transformed data.

As the main aim of the study was to determine whether elevated temperatures would significantly influence dGC levels in the species, we opted for a s linear mixed model ('lme4' package in R; *Bates et al., 2015*) looking at the effect of "state" (fixed effect) on log transformed dCG levels (dGC_log, response variable), with ID as random effect. *P*-values were calculated using the 'lmerTest' function (*Kuznetsova, Brockhoff & Christensen, 2017*). In the event of statistical significance, the 'emmeans' (*Lenth, 2023*) function was used to estimate marginal means for each stage, and pairwise comparisons were adjusted for multiple testing using Tukey's method.

### Weight

Mass change was measured for each participant. The mass change was defined as the difference in mass from baseline to the end of the intervention period. A linear mixed model was used to analyze the effect of group (Experimental *vs.* Control) on mass change. The model accounted for random effects of participants to control for individual variability. Mass change was the dependent variable, group the independent variable (fixed effect) and (1 |ID) the indicating the random effect of study animals. The fixed effect included in the model was the group. The analysis was performed using the lme4 (*Bates et al., 2015*) package in R.

## RESULTS

The results of the linear mixed model indicated that the "Elevated" state was significantly higher than the reference state ($z = 3.299$, $p = 0.001$, Table 1). The model included an intercept for individual, reflecting the variability in baseline dGC_log levels across individual animals. In addition to this, the model provided a good fit for the dGC_log data used, as evident by the log-likelihood value of 155.043. The subsequent post-hoc analysis conducted in response to the model findings showed that there was a significant difference in dGC_log levels between the "Elevated" state and all other states used ($p < 0.001$, Table 1). The largest difference in Dgc_log levels was observed between the baselineEXP

**Table 1  The statistical results of the simplified linear model and *post-hoc* analyses.**

| Model summary | | | | | |
|---|---|---|---|---|---|
| **Term** | **Coefficient** | **Standard error** | **z-value** | **P-value** | **95% Confidence Interval** |
| Intercept | −0.707 | 0.016 | −43366 | <0.001 | (−0.738, −0.675) |
| state: BaselineCtr2 | 0.004 | 0.017 | 0.247 | 0.805 | (−0.029, 0.037) |
| state: BaselineEXP | −0.029 | 0.023 | −1.258 | 0.208 | (−0.074, 0.016) |
| state: Elevated | 0.076 | 0.023 | 3.299 | 0.001* | (0.031, 0.121) |

| Tukey HSD *post-hoc* analysis | | | |
|---|---|---|---|
| **Comparison** | **Mean difference** | **P-value** | **Significant difference?** |
| BaselineCtr *vs.* BaselineCtr2 | 0.0041 | 0.9957 | No |
| BaselineCtr *vs.* BaselineEXP | −0.029 | 0.3785 | No |
| BaselineCtr *vs.* Elevated | 0.076 | 0.0003* | Yes |
| BaselineCtr2 *vs.* BaselineEXP | −0.0331 | 0.2612 | No |
| BaselineCtr2 *vs.* Elevated | 0.0719 | 0.0006* | Yes |
| BaselineEXP *vs.* Elevated | 0.105 | <0.001* | Yes |

**Notes.**

Model summary and *post-hoc* analyses results for the linear mixed effect model. For the model, 'log transformed dermal glucocorticoid levels' was set as the response variable, while 'state' was the fixed effect. 'ID' was included as a random effect to account for individual differences.

Significance is indicated by *.

and Elevated periods (Table 1). The graphical analysis supported the findings of the linear mixed model (Fig. 1). While the median values for the "Elevated" state was significantly higher than all other states, it should be noted that the standard deviation observed was considerable for all states ("BaselineCtr" = 0.075, "Baseline Ctr 2" = 0.085, "BaselineEXP" = 0.064. "Elevated" = 0.077; Fig. 1); such a high degree of deviation indicates a high level of individual variation within each state.

The linear mixed model analysis revealed a significant effect of group on mass change ($z = −6.96$, $p < 0.001$). The fixed effect of group indicated that the experimental group (average change: −9.89%, Stdev = 4.02, Fig. 2) had a significantly different mass change compared to the control group (average change: 2.56%, Stdev = 3,78). The variance of the random effect was estimated to be 7.618, indicating individual variability in percentage mass change.

# DISCUSSION

This study clearly demonstrates the effect of elevated temperature on the physiological response of *P. edulis*, including significant increases in observed dGC secretion and a notable decrease in individual weight. More importantly, this research highlighted the usefulness of using dGC monitoring to determine the effect of environmental temperature on amphibian welfare.

The experimental group of this study showed a significant increase in dGC concentrations due to higher environmental temperatures. Such temperature increases

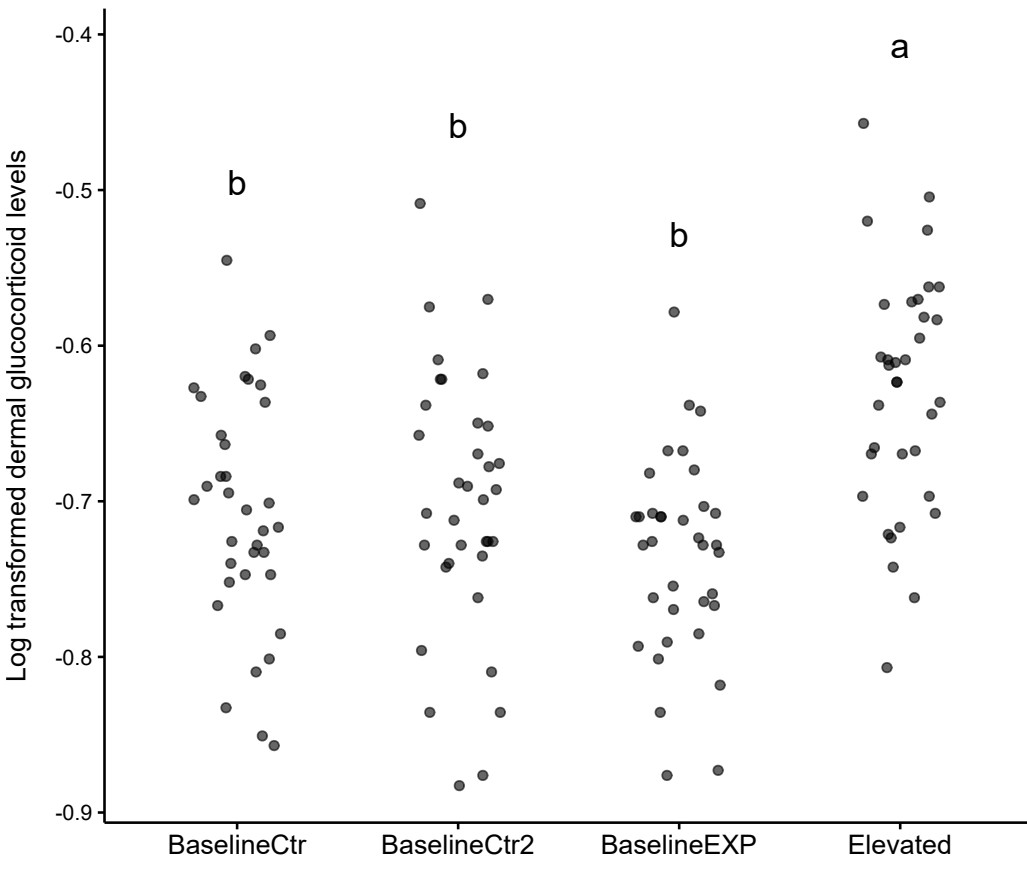

**Figure 1** **Dotplot of log transformed dermal glucocorticoid levels across different states.** Dot plot depicting log transformed dermal glucocorticoid levels in four experimental states: BaselineCtr, BaselineCtr2, BaselineEXP, and Elevated. Each dot represents an individual measurement. Significant differences are denoted by the letters above each state (a,b).

have consistently shown adverse impacts on ectotherms, highlighting a universal stress response across diverse species. Notably, research on various reptile species, including the garter snake, *Thamnophis elegans* (*Gangloff et al., 2016*), and the northern alligator, *Elgaria coerulea* (*Telemeco & Addis, 2014*), support these findings. This hyperactivation of the HPI axis in response to elevated temperature has also been observed in several amphibian species, such as the red-backed salamander, *Plethodon cinereu* s (*Novarro et al., 2018*), but also in in the cane toad, *Rhinella marina* (*Jessop et al., 2018*; *Narayan & Hero, 2014*). As such, the current study aligns with broader findings that illustrate an increase in GC secretion as a common physiological response to thermal stressors among ectothermic taxa. The level of dGC increase observed in the current study is similar to that found during an ACTH-challenge conducted on the species (*Scheun et al., 2019*). This further supports the physiological sensitivity of amphibians to environmental temperatures (*Duarte et al., 2012*; *Novarro et al., 2018*).

Although the dGC levels of control animals were significantly lower than experimental animals, several individuals showed higher than expected dGC concentration during

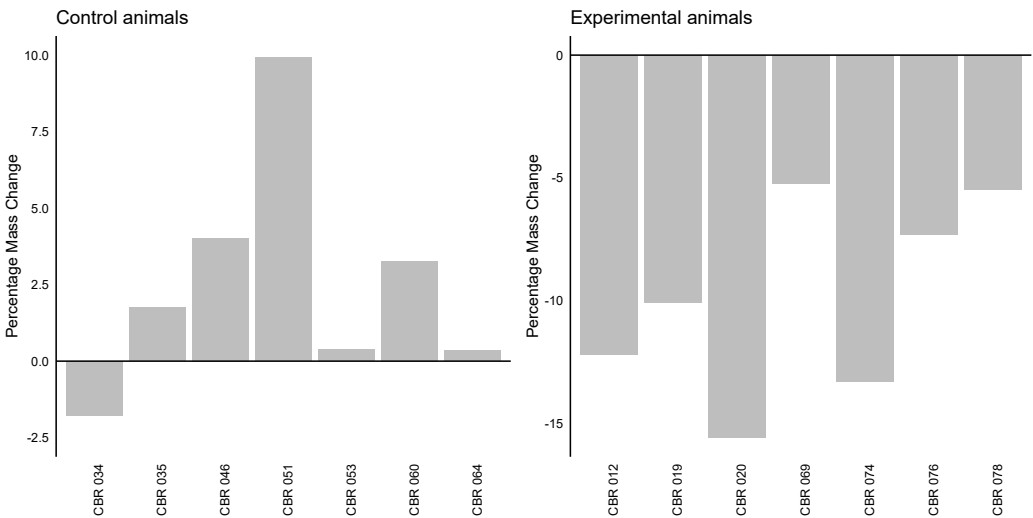

**Figure 2** **Percentage mass change in control and experimental animals.** A bar graph indicating the percentage mass change (in percentage) for individual control and experimental animals. The horizontal line at y axis point 0 indicates baseline values for all individuals.

Baseline 2. Research has demonstrated the presence of variations in both the physiological stress response and baseline levels of glucocorticoids among individuals and between different sexes (*Bourke, Harrell & Neigh, 2012*; *Eliason et al., 2020*; *Koolhaas et al., 2010*). Furthermore, capture events might have resulted in elevated dGC levels as the study proceeded (*Meijer, Sommer & Spriuijt, 2007*; *Narayan, Cockrem & Hero, 2013*). However, additional research needs to be conducted to determine which of the discussed factors might have been responsible for the elevated dGC levels in certain animals of the control group.

All animals within the experimental group showed a significant decrease in weight compared to the control animals. Ectotherm metabolism, like many aspects of their physiology, has evolved under specific environmental temperatures to ensure optimal physiological function (*Navas, Gomes & Carvalho, 2008*). The use of an external energy source to regulate internal temperatures, though energetically efficient, does make ectotherms vulnerable to any change in environmental temperatures (*Li, Cohen & Rohr, 2013*). An increase in environmental temperature will lead to greater energetic demands that will likely exceed the resources available to amphibians (*Rollins-Smith & Le Sage, 2023*). This might well explain the loss in body weight of the experimental animals; as experimental temperatures were increased during the study, food availability remained constant, creating a deficit between energy requirements and availability. While this deficit is easily removed within the captive environment, free-ranging amphibians will have to increase their foraging efforts to meet daily energy requirements (*Rohr & Palmer, 2013*). Engaging in further activity is expected to further increase body temperature and exacerbate water loss across numerous species, thereby intensifying the issue (*Li, Cohen & Rohr, 2013*; *Navas, Gomes & Carvalho, 2008*). This clearly highlights the complexity of

the threat climate change poses to free-ranging individuals. A holistic approach should be considered when studying the effect of climate change on any amphibian species, with aspects such as behavioural and physiological responses included in the analysis.

Another aspect to consider when assessing the physiological stress and weight loss in our study animal, is the level of evaporative water loss (EWL) that might occur. Unlike reptiles, amphibians have highly permeable skin that they use to control their internal water balance, but also to assist in regulating body temperature when environmental temperatures exceed the thermoneutral zone (*Bickford et al., 2010*; *Lertzman-Lepofsky et al., 2020*; *Sannolo, Barroso & Carretero, 2018*). As water balance maintenance is crucial for growth, performance, and survival within amphibian species, EWL is often employed by amphibians to survive environmental extremes (*Peterman, Locke & Semlitsch, 2013*). However, recent climate change models covering large regions of eastern and southern Africa, in which the current study species can be found, predict an increase in average annual temperatures, while fluctuations in precipitation is also expected (*Hulme et al., 2001*; *Nyong & Niang-Diop, 2006*). Such extreme or prolonged events will result in considerable EWL in the majority of amphibian species (*Lertzman-Lepofsky et al., 2020*). This will in turn result in a loss in weight as well as an increase in physiological stress experienced. For captive amphibians such as *P. edulis* this might not be a lethal aspect to overcome. However, considerable EWL in free-ranging amphibian populations, and the inability to meet the resulting water requirements, may result in several lethal and sublethal effect (*Blaustein et al., 2010*). However, to further enhance our understanding of EWL in the study species, additional research on captive and free-ranging populations are required.

While the strain produced by both elevated metabolic rates and EWL may have been responsible for the observed increase in dGC concentrations and the decrease in weight of experimental animals, it must be noted that a complex relationship exists between temperature, metabolic rate, water regulations and the HPI axis. Several studies have shown the role of GC in driving metabolic rates in mammalian and herptile species (*Crespi & Denver, 2005*; *De Bruijn & Romero, 2018*; *Wack et al., 2012*). Hence, the observed rise in GC levels due to heightened ambient temperatures could potentially have led to an increase in metabolic activity, resulting in a discernible loss of mass among the animals in the experimental group. Similarly, dehydration in amphibians will increase HPI-responsiveness, leading to an increase in secreted GCs (*Madelaire et al., 2020*). As such, an increase in environmental temperature, a decrease in water availability and higher levels of EWL are responsible for elevated GC levels, which in turn will affect metabolic rates. This complexity, along with the behavioural response inherent in each species, must be considered when conducting climate change related research on captive and free-ranging populations.

## Mucous as a matrix for monitoring glucocorticoid levels in amphibians

An important finding of this research is the capability to track GC fluctuations in response to temperature changes within the species by using mucous as a biological matrix. While both urine and faeces are robust matrices to monitor HPI activity in

amphibians (*Narayan et al., 2019*; *Santymire, Manjerovic & Sacerdote-Velat, 2018*), both have considerable shortcomings. Urine collection often requires considerable handling of the animal, including massaging of the underbelly (*Narayan, 2013*), while also increasing dehydration of individuals. Moreover, obtaining fresh fecal samples from free-ranging individuals presents a considerable challenge due to the difficulty in observing defecation events. Additionally, researchers may need to confine individuals in containers until defecation takes place to guarantee the collection of a sample (*Schilling, Mazzamuto & Romeo, 2022*). In contrast to both matrices, mucous can be collected by briefly swabbing an individual whenever encountered; these individuals can then be released immediately, thus minimizing handling stress and waiting times. Implementing this method within captive settings would enable managers to efficiently evaluate HPI axis activity, as well as identify potential deficiencies in the captive environments. This facilitates the adoption of optimal animal management practices to promote the well-being of the inhabitants. Furthermore, the simplicity and semi-invasiveness of mucous sampling establish it as an ideal tool for monitoring GC levels in free-ranging populations. This method allows for the efficient assessment of environmental stressors and their impacts on population survival throughout the distribution range of *P. edulis*.

## Shortcomings and future research

The results of this study clearly indicate a considerable physiological response of *P. edulis* to elevated environmental temperatures, as observed in a significant increase in secreted GC levels, while experimental animals also experienced a considerable loss in weight. These findings support similar studies indicating the sensitivity of amphibians to temperature changes (*Jessop et al., 2018*; *Narayan & Hero, 2014*; *Novarro et al., 2018*). However, this study should be seen as another piece of the puzzle in understanding the responses of amphibians to climate change. As such, there are several areas that can be improved upon to enhance our understanding. Firstly, the setup of this study focused on the direct effect of temperature on the physiological stress response; in doing so, the environment used was simplified, with constant food and water availability. As climate change will result in a decrease in food availability (*Sangle et al., 2015*) as well as unpredictable rainfall patterns (*Dore, 2005*), it is of utmost importance that studies include a holistic approach, including water unpredictability and food shortages, but also setups that can describe the link between temperature, EWL and the physiological response. Furthermore, though sensitive to temperature changes, amphibians have also developed a range of behavioural adaptations to assist in surviving these extreme periods (*Bodensteiner et al., 2021*; *Enriquez-Urzelai et al., 2020*). The setup employed in this study minimized the ability of both experimental and control animals to employ the full range of behaviours available; this might have contributed to elevated GC levels observed in both groups. Future studies must prioritize experimental setups that permit subjects to exhibit innate behaviors, such as burrowing, to ensure ecological validity. While controlled experiments provide a valuable framework to scrutinize specific facets of amphibian responses to climate change, there's a need for continuous monitoring of free-ranging populations. Integrating long-term observational studies is crucial for capturing authentic responses of these populations to environmental

shifts, thereby enriching our understanding of their adaptability and resilience in the face of climate change.

## CONCLUSIONS

While the impacts of extreme environmental conditions on amphibians are widely recognized, there is limited understanding of the direct physiological and physical effects, particularly concerning African species. This study was conducted to rectify this limitation, focusing on a prevalent African amphibian species and employing mucous as a matrix. While an increase in secreted GCs was expected, the scale thereof was surprising. Peak dGC levels not only aligned with prior ACTH results from the species but also highlighted a lack of decline throughout the experiment, underscoring the inability of the study species to adapt and/or acclimatize to extreme conditions. While the 5 °C increase used during this study seems excessive, it does stress the response of the edible bullfrog under extreme, long-term environmental changes. Furthermore, it suggests that beyond a specific threshold, amphibians struggle to regulate biochemical processes, even in the presence of a consistent supply of food, water, and shelter. In such instances, merely improving the habitat for endangered species grappling with climate change will prove inadequate for the effective conservation of their populations in the wild. The findings of this study contribute to our understanding of how African amphibians could react to climate change; however, it should be regarded as a preliminary step, encouraging researchers to undertake more intricate investigations involving this species and others. Research must be conducted to investigate the significance of factors such as temperature, humidity, water and food availability, as well as their combined effects, on the physiological responses in amphibians.

## ACKNOWLEDGEMENTS

We would like to thank the staff of the Reptile and Amphibian Section at the NZG for their assistance throughout the study. We would also wish to thank the technicians at the Endocrine Research Laboratory for the sample analyses.

### Funding
The authors received no funding for this work.

### Competing Interests
The authors declare there are no competing interests.

### Author Contributions
- Juan Scheun conceived and designed the experiments, performed the experiments, analyzed the data, prepared figures and/or tables, authored or reviewed drafts of the article, and approved the final draft.

- Leanne Venter conceived and designed the experiments, analyzed the data, prepared figures and/or tables, authored or reviewed drafts of the article, and approved the final draft.
- Andre Ganswindt conceived and designed the experiments, analyzed the data, authored or reviewed drafts of the article, enzyme immunoassay analyses, and approved the final draft.

## Animal Ethics

The following information was supplied relating to ethical approvals (*i.e.*, approving body and any reference numbers):

Research approved by the SANBI National Zoological Garden Animal Use and Care Committee (Reference: P19/15)

## Data Availability

The raw data are available in the Supplemental Files.

## Supplemental Information

Supplemental information for this article can be found online at http://dx.doi.org/10.7717/peerj.17847#supplemental-information.

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
