# Peer review of "A frog in hot water: the effect of temperature elevation on the adrenal stress response of an African amphibian"

_PeerJ, doi:10.7717/peerj.17847_

## Round 0.1 · original submission · Major Revisions

Thank you for submitting you manuscript to PeerJ. Your paper has been reviewed by two experts in the field and both have offered valuable and constructive feedback on your manuscript.

I agree with the reviewers about the lack of details in the methods, particularly the statistical analysis section. To add to the comments by reviewer 2 (and I strongly agree with their points 1 and 2 specifcially about the statistical analyses):
You comment that you log-transformed the data to achieve normality, but then ran a GLMM - the term "generalized" in statistical language means that the data did not fit the assumptions for a linear model, which uses a normal distribution. If you actually ran a GLMM, then what distribution did you use? - you used lme4, I assume the function glmer() - but if you didn't specify the distribution then it uses a Gaussian distribution (which is the normal distribution), and you could do the same thing using the function lmer().

I agree that the AIC model ranking is a little confusing because you don't describe enough variables to warrant this. is AIC just to check if there is an interactive effect? As this is a laboratory study, where you controlled for confounding effects, could do one of two things. Just run the model with the interactive effect of group*time, and report the results. Or if you don't think that's the correct option, say clearly that the model selection is to test if the interactive effect is important, and if its not, then you report the additive model. This way you can use the package 'car' and the function Anova() and get the model results, and you do not have to do all the extra work in lines 193-194, which is not appropriate for the relatively simple model you are conducting here.

I also agree that the figures are not easy to read, which makes understanding the overall patterns difficult. For figure 3, because all the animals were different sizes, finging the patterns of change is confusing, and might be better represented as a change in size from one time point to the next. The figures (all of them) focus on the effects in the individuals, but as a a planned intervention study with a reasonable sample size, you should be focusing on the change/effect overall.

In your revision, please respond to each point made and make the associated changes directly in the manuscript. I look forward to reading your revised draft.

Reviewer 1 ·

Basic reporting

The novelty and scientific importance of the work are notable, but I believe that further exploration of the scientific literature is necessary, primarily to support the choice of the specie used as a model. Furthermore, although the objective of the manuscript is clear, it would be crucial to explicitly state the hypotheses behind it.

The figures are relevant to the content of the manuscript, but have flaws such as labeling, standardization of terms used, and absence of axis lines. All faults are detailed in the attached revised material.

According to PeerJ guidelines, coherent bodies of work should not be subdivided inappropriately just to increase the number of publications. I am confused by the mucous extraction protocol, cited as "Scheun J. 2024. 2024 PeerJ Submission: Mucous Extraction. Protocols.io, 2024. DOI:dx.doi.org/10.17504/protocols.io.14egn3qrzl5d/v1". I believe that this material is part of the experiment carried out for the manuscript in question and should be included in it, either in the main text or in supplementary material, and not as a new publication, if applicable.

All details of my suggestions are described and expressed in comments in the attached revised material.

Experimental design

The methods described lack sufficient detail and information to be replicated. Comments and suggestions for improving the detailing and making the method clearer are outlined in the attached revised material.

Validity of the findings

All underlying data have been provided, are statistically robust, and controlled for by the low sample size and pseudoreplicates. The conclusions are well-formulated, but there is a disparity in the approach of topics between the introduction and discussion, and conclusion.

All details of my suggestions are described and expressed in comments in the attached revised material.

Annotated reviews are not available for download in order to protect the identity of reviewers who chose to remain anonymous.

Reviewer 2 ·

Basic reporting

1) I recommend using simpler statistical methods. You are ultimately interested in whether the effect of treatment (control vs elevated) influences dGC. To get at this question, I would use a linear mixed model with an interaction term of treatment group*time period [e.g. log(dGC) ~ group*period + (1|animal)] and then use post-hoc analysis (with a package like multcomp [https://cran.r-project.org/web/packages/multcomp/vignettes/multcomp-examples.pdf] or emmeans [https://cran.r-project.org/web/packages/emmeans/vignettes/interactions.html#contrasts]) to look at the contrasts of interest (e.g. control group – Baseline 1 and Baseline 2 will likely not be different from one another, but treatment group – Baseline 1 and Elevated will be different – basically, the effect of time period will depend on treatment). Alternatively, please describe your current statistical methods in more detail. I’ve included comments below on specific areas in the statistical methods that were confusing to me but overall, the paragraph on the data analysis was not clear to me.
2) I recommend using different data visualizations to better show the effects of elevated temp on dGC and body mass: Fig 1: Since you are plotting mean points here, I would include SE bars around the points so readers can assess the amount of variability within each time point. Since your analyses are on log-transformed data, you might consider also plotting the data on a log-transformed scale rather than the raw data. Fig 2: Why/how did you pick 2 animals to plot from each treatment group? One option would be to include individual trajectories for each animal with a line for the group average overlayed on top. Please include an x-axis title (week). Fig 3: A boxplot showing the difference in the change in weight/mass between treatment groups would be more intuitive for a reader (x-axis = treatment group (control vs elevated), y-axis = change in mass). You could have one plot for the change from week 1 to week 5 (no difference between groups) and another plot for the change from week 5 to week 10 (decrease in elevated temp, no change for control) with individual points overlayed on the boxplot for each individual animal.
3) I would appreciate more context as to why you chose 30C as the elevated temperature. Is this temperature within their natural range? Above the natural range? Based on your description of the housing conditions, the frogs were unable to behaviourally respond to the increase in temperature by burrowing underground as they would naturally do. This species naturally spends most of the year in aestivation underground. Given this behaviour, how is this increase in temperature relevant to this species in particular? Please add some species specific context to the introduction or discussion.


Minor edits:

Line 60-61: I recommend adding an estimated total species # for context, given 110 species is only a small fraction of the total number of amphibian species (~8800 sp.)

Line 75: I recommend changing the wording slightly – GCs can alter behaviour, but they don’t necessarily “enhance” behaviour

Line 76-77: This is minor but “chronically elevated GC levels can be detrimental” to all vertebrates, not just amphibians. There are many more citations you can lean on here to show the detrimental effects across vertebrates to better motivate why your study is needed in amphibians.

Lines 83-110: Fecal samples provide an integrative measure of GCs across a time period whereas plasma samples are considered more instantaneous samples allowing you to assess GCs at a moment in time. Where do mucous samples fall in regard to interpreting GCs? Are they more integrative or instantaneous?

Line 103: I believe “dibble” is a typo for “edible”

Line 106: When I read “sex determination”, I interpret it as the process that determines biological sex or the development of biological sex, rather than what I think you mean here: the identification of sex. Please rephrase.

Line 114-116: I recommend clarifying what exactly you mean by temperature change in this sentence, i.e. change in what direction and for how long? Also, a change in weight/mass can still be considered a physiological response – perhaps rephrase to “physiological responses through the HPI axis and body mass” or something similar.

Line 128: “within” should be “from”

Please include a brief description of the extraction protocol and the EIA, in addition to citing the papers with the full protocols, to ensure readers understand all your methods without going back to other papers.

Line 194-197: It’s not clear to me what you mean with these sentences about the global model. What interactions are you omitting? What subsets? If you’re using a GLMM, what distribution are you using?

Line 197-198: I would remove this sentence about random effects since you discuss it in line 190-191. I’m not sure what you mean by “animal (1|animal”.

Lines 205-208: It’s not clear why you used both an ANOVA and a t-test to look for a change in body weight. The results of the ANOVA were not clearly reported in the result section.

Line 247: Typo – ACTH, not ACTh

Supplemental Data – Why is frog CBR 053 highlighted in green in the dGC data?

Experimental design

Please include a brief description of the extraction protocol and the EIA, in addition to citing the papers with the full protocols, to ensure readers understand all your methods without going back to other papers.

Validity of the findings

Throughout the paper, there are multiple places where you indicate this is “the first study” or “no study to date…”. Personally, I think you should limit how often you say this point. This study adds valuable empirical data, but many other studies are doing similar work, many of which you have cited. Calling it “the first” is a significant claim in my view and doesn’t add value to the story.

---

## Round 0.2 · Minor Revisions

Thank you for submitting your revised manuscript to PeerJ. You have done an excellent job addressing the reviewers and my comments in this revision. There are a few additional points that should be addressed, please see the comments from Reviewer 2.

Thank you and I am looking forward to your resubmission.

Reviewer 1 ·

Basic reporting

After carefully reviewing the paper titled "A frog in hot water: the effect of temperature elevation on the adrenal stress response of an African amphibian" in its second round of review, I would like to express my satisfaction with the substantial changes implemented by the authors in response to the comments and suggestions provided in the previous round.

The authors demonstrated a remarkable commitment to improving their work by comprehensively addressing all concerns raised by reviewers. Their modifications resulted in a more robust, detailed, and, crucially, more convincing paper. The questions raised regarding the methodology have been carefully addressed, now providing a solid basis for the results presented. Additionally, the discussion has been expanded and deepened, providing a more complete analysis of the results and their implications. Overall, I am convinced that the changes made by the authors resulted in a substantial improvement in the quality of the paper.

Experimental design

No comment.

Validity of the findings

No comment.

Reviewer 2 ·

Basic reporting

The authors have sufficiently addressed the additional context I suggested adding to the manuscript and have done a clear job in reanalyzing the data with a more simplified model structure for the dGC data.

Experimental design

It is clearly stated how this experiment fills an identified knowledge gap. The author's revisions to the methods have clarified them and provide enough information to replicate.

Validity of the findings

One minor comment about the new statistical analysis section is to include mention of the random effect of individual ID.

I was confused why the y-axis for Figure 1 was only negative values (indicating concentrations were <1). However, when I downloaded the dGC data, it was not clear which headers belonged to which columns and I couldn't tell if it was the raw dGC data or transformed so I couldn't verify the underlying data for those analyses. I recommend the authors take another look at the CSV file they provided for the dGC data.

I suggest the authors try a different analysis for the body mass data using “change in body mass” as the response variable rather than weight at that time point. Given the apparent difference in weight loss between the treatment groups but the lack of statistical significance, it may be worth including a power analysis to find out what sample size would be needed for this effect to be statistically significant. I think the effect is real but your sample size is quite small and limits your ability to find a subtle effect.

As was mentioned in the discussion, I think it’s difficult to disentangle whether this change in body mass is rather a change in evaporative water loss (and thus a change in body water content) or an effect of an increased metabolism or a combination of the two. I appreciated the inclusion of these potential hypotheses in the discussion.

Additional comments

Figure 2B: Instead of plotting the weight at each time point, I recommend plotting the “change in mass” as suggested by the editor. I have included code for this in the attachment.

Annotated reviews are not available for download in order to protect the identity of reviewers who chose to remain anonymous.

---

## Round 0.3 · Minor Revisions

I did receive your email, and this manuscript didn't go out for additional review, I was just on leave for 4 weeks, and therefore must apologise for the delay.

I think you've done a great job addressing the reviewer questions, for the most part.

I have a couple of points that would be great if you can address them before we move to publish, and they are about the readability and utility of your figures, and the appropriateness of one of your analyses.

I agree with the reviewer about reporting change in mass rather than raw mass for 2A. Your statistical test reports that there is no effect of week or mass of the animal (line 294). Here, there are some issues with the correctness in your stats and you statements that need to be addressed.
1) These are repeated measures, which means a Kruskal-wallis test is not appropriate, as repeated measures violates the assumption of data independence. You should be running a mixed effects model. I haven't played around with you specific data, but you should be able to log transform mass, and run an Linear mixed effects (LME) model where ID is a random effect. This would be more appropriate. If there is an effect, you might be able to see if better in the graph with a log-transformed y-axis.
3) do an LME but the response variable is change in mass (2 data points, change in mass from week 1-5, and week 5-10 (or 1-10 if that makes more sense to you). The mass of these animals are different to start, therefore the lines across in your figure are mostly flat, although there is perhaps a dip in some of the animals at week 10. If you reported change in mass, it would level out the initial weight differences at the start and show the patterns in mass loss.
3) you need to report the results of the stats as is. If the stats say there is no effect, then you need to report that there is no effect. Don't include a figure because there is no effect. The figure you have presented, to me, looks like the lines are mostly flat, so it doesn't look like there is an effect of time on mass anyway. and delete lines 294-298 and the figure, because they are not supported by the data and the analyses.

Figure 2B, what is this supposed to show? is it different than 2A? to me, they both support that there is no effect on mass, so there is no reason to have either figure, although the stats do need to be redone with a more appropriate model.

Figure 1: please add the individual data points to the boxplots. and use notation in the figure to indicate which group is statistically different from the rest, with a,b,c notation is probably the easiest way to visually represent the Tukey HSD outputs from table 1

Table 1: clarify the model used (what do you mean by simplified LME? what is the response variable? what are the fixed and random effects?)

---

## Round 0.4 · accepted · Accept

Thank you for making these edits, the appropriate model has given you clearer results, and I believe the manuscript is now ready for publication.

Congrats! I'm looking forward to seeing it in print!